



# Advantages of G-band radar in multi-frequency, liquid phase microphysical retrievals

Benjamin M. Courtier[1], Alessandro Battaglia[2], and Kamil Mroz[1,3]

[1]University of Leicester, Leicester, UK
[2]Politecnico di Torino, Turin, Italy
[3]NCEO, University of Leicester, Leicester, UK

**Correspondence:** Benjamin M. Courtier (bmc19@leicester.ac.uk)

**Abstract.** Radar based microphysical retrievals of cloud and droplet properties are vital for informing model parameterisations of clouds and precipitation but these retrievals often do not capture the details of small droplets in light rain or drizzle. A state-of-the-art G-band radar is used here to demonstrate improvements to microphysical retrievals in a case study featuring light rain. Improvements are seen, as compared to W-band radar, in the retrieval of vertical wind speed, due to the location of Mie

minima at smaller droplet sizes with the G-band radar. This, in turn, has an impact on the retrieval of the drop size distribution, allowing for better accuracy in the retrieval of the characteristic drop diameter and improvements in the retrieval of the number of concentration of small droplet sizes. The differential Doppler velocity between Ka- and G-bands shows increased dynamic range compared to the Ka-W pairing, particularly for instances presenting small characteristic drop diameters. The increased attenuation experienced at G-band enables improved retrievals of liquid water content and precipitation rate when paired with

W-band or Ka-band as compared to the W-band and Ka- band pairing. This is particularly noticeable in periods of light rain where the W-band and Ka-band radars receive negligible attenuation while the attenuation at G-band is much greater.

## 1 Introduction

Radar based microphysical retrievals of cloud and droplet properties are vital for informing model parameterisations of clouds and precipitation. In-situ measurements of microphysical parameters are very informative, but do not have the breadth of

coverage both geographically and in terms of the range of cloud types that can be sampled by using radars and in particular satellite based radars (e.g. Kidd, 2001; Mason et al., 2023). As improvements in solid-state radar technology have fostered the development of the next generation of meteorological radars in the G-band region (Cooper et al., 2018; Lamer et al., 2020; Courtier et al., 2022), the potential for a satellite-borne G-band radar is of great interest for improving the retrievals of the smallest particles found in both ice and liquid clouds (Battaglia et al., 2020).

Current rain microphysical retrievals typically use W-band, Ka-band, X-band or a combination of any or all of those frequencies (e.g. Tridon and Battaglia, 2015; Tridon et al., 2020; Mróz et al., 2021; Von Terzi et al., 2022). This is appropriate for many atmospheric conditions and works well for rain and raindrops exceeding $\sim 0.5\,\mathrm{mm}$. But most cloud droplets are too small to be effectively observed using these radar frequencies, assuming standard radar sensitivities. Small cloud droplets are observed by all of X-, Ka- and W-band radars in the Rayleigh regime, meaning that no size information can be diagnosed from





multi-frequency techniques. G-band radar is predicted to improve the retrievals of microphysical properties such as LWC or rainfall rate for these small sized droplets or particles (Mead et al., 1989; Lhermitte, 1990)

The microphysical retrievals that are of interest to this study are: the vertical wind, the Drop Size Distribution (DSD), the characteristic droplet diameter, the Liquid Water Content (LWC, which is related to the Path Integrated Attenuation, PIA). These are introduced in more detail below. Battaglia et al. (2014) detailed the improvements that can be made in these retrievals

by using a G-band radar in combination with other cloud radars. They state that the vertical wind can be observed with median volume droplet diameters ($D_0$) as small as $0.23\,\mathrm{mm}$ (with a spectral broadening of less than $0.2\,\mathrm{m\,s^{-1}}$). This is a substantial improvement over the smallest $D_0$ possible to retrieve the vertical wind in at W-band. The improvement in the LWC is largely due to the increased differential attenuation observed by the inclusion of a G-band radar, Battaglia et al. (2014) suggest a four fold improvement in accuracy for a Ka-G pairing as compared to a Ka-W pairing.

**1.1   DSD retrieval**

DSD retrievals vary greatly in terms of complexity, the most simple of which is to assume no turbulence in the atmosphere and just retrieve the DSD by the spectral power in a velocity bin. An efficient method of Williams et al. (2016) assumed the DSD could be represented by a gamma distribution and used Doppler velocity difference (the difference between mean Doppler velocity at two difference wavelengths) to retrieve the parameters of the gamma distribution and therefore the DSD.

Many methods use variational techniques to retrieve the DSD; for example Mason et al. (2017) use the moments of airborne, Doppler radar observations together with the PIA to retrieve parameters to produce a gamma based DSD with a fixed shape parameter. More complex methods (Firda et al., 1999; Tridon et al., 2017; Mróz et al., 2021) use observations of Doppler spectra to adjust binned retrievals of DSD using turbulence and the vertical wind to adjust the shape, and path integrated attenuation to adjust the magnitude of the spectra. This was done using a simple iterative method by Firda et al. (1999) and

using an optimal estimation technique by Tridon et al. (2017) and Mróz et al. (2021).

**1.2   DDV-based retrieval**

DDV retrievals aim to retrieve a characteristic diameter of the DSD without retrieving the entire DSD; this can then be used to estimate the full DSD if some assumptions are made (e.g. Tian et al., 2007). This is advantageous if the full Doppler spectra is not recorded and instead only the moments are known. The method relates the differential Doppler velocity (which

is independent of the vertical wind) to, typically, the mass-mean weighted diameter ($D_m$). This relationship is derived via a statistical, observational approach (e.g. Matrosov, 2017) or a theoretical one (e.g. Tian et al., 2007).

Using only two radar wavelengths has the issue that there may be double solutions for each DDV observation; often this can be resolved using the Doppler velocity or reflectivity to discriminate the right solution based on the fact that larger particles are associated to larger signals in such variables. However, there can still be ambiguous results, particularly for values of $D_m$

where the two solutions are similar. Mróz et al. (2020) presented a triple-frequency retrieval in which the uncertainty in the retrieval of $D_m$ is reduced for large $D_m$. They also suggested that, in order to improve the ability to retrieve small $D_m$s, the inclusion of a G-band radar is necessary.



### 1.3 PIA-based retrieval

The path integrated attenuation is difficult to retrieve because of the entanglement of attenuation and non-Rayleigh effects which both affect the radar reflectivity (Tridon et al., 2020). If multiple frequency band radars are being used it becomes slightly easier to separate attenuation and non-Rayleigh scattering. One method to disentangle these effects, implemented by Tridon et al. (2013), aligns the Rayleigh regions of the spectra and uses the reflectivity adjustment to estimate the differential attenuation. If one of the radar frequencies used can be approximated as receiving no attenuation then the absolute attenuation can also be retrieved. Attenuation is caused by both gaseous and hydrometeor attenuation. If the gaseous attenuation is known (this can be simply calculated using a radiosonde profile or reanalysis model data) then any remaining attenuation below the freezing level is caused by liquid water. Hogan et al. (2005) demonstrated a method for retrieving LWC using the fact that differential attenuation is proportional to liquid water content in a cloud.

These three retrieval techniques are used throughout this study to exemplify the benefits of the G-band radar when retrieving microphysical properties. The retrieval of the full DSD is of particular importance as the other microphysical properties of interest can all also be calculated if the full DSD is known.

In this study we detail the theoretical advantages of G-band radar with respect to microphysical retrievals and then examine the actual performance of the G-band radar based in Chilbolton in retrieving microphysical properties. In Section 2 we present an overview of the theoretical performance of a G-band radar for several different retrievals. In Section 3 we describe the retrieval methods used and the case study data that is used to verify the retrievals. In Section 4 we demonstrate the real life capabilities of a G-band radar with regards to the microphysical retrievals and relate the performance back to what is expected from theory. In Section 5 we summarise the study and present the capabilities of G-band radar.

### 2 Theory

The main advantages of G-band over lower frequencies are in the interactions between the $1.5\,\mathrm{mm}$ radar transmitted electromagnetic wave and droplets of a similar size to this radiation. At $200\,\mathrm{GHz}$ the microwave radiation scatters in the Rayleigh regime (i.e. when the backscattering cross section differs from the Rayleigh counterpart by less than $3\,\mathrm{dB}$) for droplets with a diameter of less than approximately $0.37\,\mathrm{mm}$; beyond this size the scattering enters the non-Rayleigh regime and the backscattering cross section is reduced as compared to the Rayleigh counterpart. Local scattering minima occur in the non-Rayleigh regime, known as Mie notches; these minima occur repeatedly until geometric scattering is reached with the pertinent notches for G-band located at droplets of diameter $0.8\,\mathrm{mm}$ and $1.5\,\mathrm{mm}$. For a vertically pointing radar observing spherical droplets falling at terminal velocity in an absence of any vertical wind at $1000\,\mathrm{hPa}$ these droplet diameters correspond directly to velocities of $3.24\,\mathrm{m\,s^{-1}}$ and $5.02\,\mathrm{m\,s^{-1}}$ respectively. Any change from this theoretical Doppler velocity is largely due to atmospheric forcing from the vertical wind (Lhermitte, 1988; Kollias et al., 1999). This technique has been used successfully by Giangrande et al. (2010) with a W-band radar to measure the vertical velocity in stratiform precipitation in Oklahoma. Because





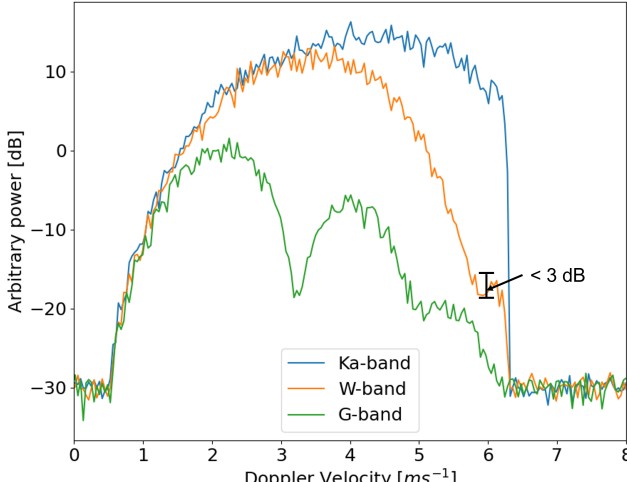

**Figure 1.** Triple frequency spectra generated from a gamma distribution with added noise. The first Mie notch in the G-band curve is clear while the second has no local minimum. The Mie notch at W-band can be seen, but does not meet the required 3 dB prominence and so would not be detected as a Mie notch.

the Mie notches occur at smaller droplet diameters for G-band radar than they do for lower frequencies the vertical wind can be retrieved at smaller rainfall rates and smaller mean drop diameters than if using longer wavelength radars.

The retrieval of the Mie notch location is done computationally using a local minimum detection algorithm with a $3\,\mathrm{dB}$ required prominence to filter out noise, this threshold could be set to be smaller but here the choice has been made to avoid the possibility of false detection. This is shown in Figure 1: the first G-band Mie notch, occurring at around $3.2\,\mathrm{m\,s^{-1}}$, clearly meets this threshold and so would be detected. The second G-band Mie notch does not have a clear minimum and so is not detected as a Mie notch. The W-band Mie minimum is slightly more complex; it can clearly be seen by eye to be a Mie notch, but is not detected by the algorithm because the prominence of the minimum is not sufficiently strong. This method is used in the optimal estimation described later in this study.

The dual-frequency ratio (DFR) can be a good proxy for the characteristic diameter estimation when at least one of the radars has some non-Rayleigh scattering contributing to the radar reflectivity. Figure 2 shows that, for pairings including the G-band and for exponential PSDs with a $D_m$ between 0.3 and $2\,\mathrm{mm}$ in particular, the DFR gives a clear diagnostic of the $D_m$. At a $D_m$ much greater than $2\,\mathrm{mm}$ the DFR levels out for the W-G pairs, but continues to increase for the Ka-G and Ka-W pairs. For Figure 1, where the $D_m$ is $0.75\,\mathrm{mm}$ the DFR between Ka-W is around $4\,\mathrm{dB}$, whereas the DFR between Ka-G is $17\,\mathrm{dB}$. The large difference between the radar reflectivity in the Ka-band and the G-band is manifested by the area between the Ka-band spectra and the G-band spectra. For the exponential distributions used to generate Figure 2 the onset of the non-Rayleigh regime, defined as above, is at a $D_m$ of $0.25\,\mathrm{mm}$.

Figure 3 demonstrates the enhanced capabilities of the Mie notch wind retrieval when using a G-band compared to a W-band radar. The figure shows where the first Mie notch can be detected in RR-$D_m$ space for various radar sensitivities. For a





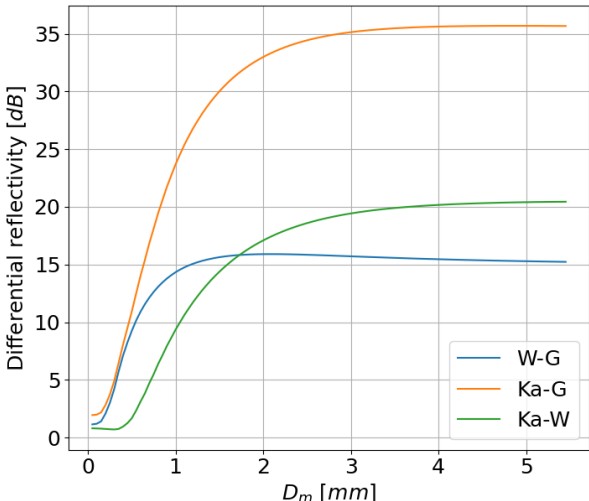

**Figure 2.** Dual-frequency ratio for three wavelength pairings for different sizes of $D_m$. All spectra were generated using exponential DSDs.

sufficiently large $D_m$ (1.3 mm) both W-band and G-band can detect a Mie notch (dark grey shaded region in Fig. 3a), although

there is a difference in the sensitivity required for each system, for the G-band, a better sensitivity is required to retrieve the
vertical wind speed as compared to W-band because of the non-Rayleigh reduction of reflectivity. However, it can be seen
in Figure 3b that for any realistic observations the sensitivity required at $D_m$s greater than 1.3 mm the sensitivity is easily
obtainable. For a sufficiently small $D_m$ (0.35 mm), neither W-band nor G-band can detect a Mie notch and therefore cannot
retrieve the vertical wind. Within the region where a G-band radar can detect a Mie notch but a W-band radar cannot (i.e. the

light grey filled region in Fig. 3a) there are small variations in the success of the G-band detecting a Mie notch.

  For the W-band simulations, there is also a dependence of the detection of the Mie notch on the shape parameter. For
narrower DSDs (i.e. larger values of $\mu$), the $D_m$ at which the Mie notch can be detected at W-band is larger than for broader
DSDs (i.e. smaller values of $\mu$). This dependence is shown in Figure 3b, for a $\mu$ of 8 (the maximum considered here) the $D_m$
required to detect the Mie notch at W-band is 1.3 mm while for a $\mu$ of 0 the $D_m$ required to detect the Mie notch 0.8 mm.

While there is a sensitivity dependence on the detection of the G-band Mie notch, for a reasonable radar sensitivity, this
only becomes relevant for very small values of $D_m$ (i.e. below 0.6 mm. At this point - for an appropriate rainfall rate, such as
$0.5 \, \text{mm} \, \text{h}^{-1}$ - the sensitivity required to detect the Mie notch is at least $-20 \, \text{dBZ}$.

  The DSD observations shown in Figure 3b display that a large number of observed precipitation events lie within the window
where solely the G-band radar can retrieve the vertical wind. If a $\mu$ of 8 is assumed (i.e. the full extent of Figure 3a) then almost

half (49.5%) the observations fall into this region. If a $\mu$ of 0 is assumed then in about 8% of the observations, the vertical wind
can be retrieved by the G-band radar but not the W-band radar. Even at the most conservative end there are a large number of
cases where only the G-band radar can provide information on the vertical wind.




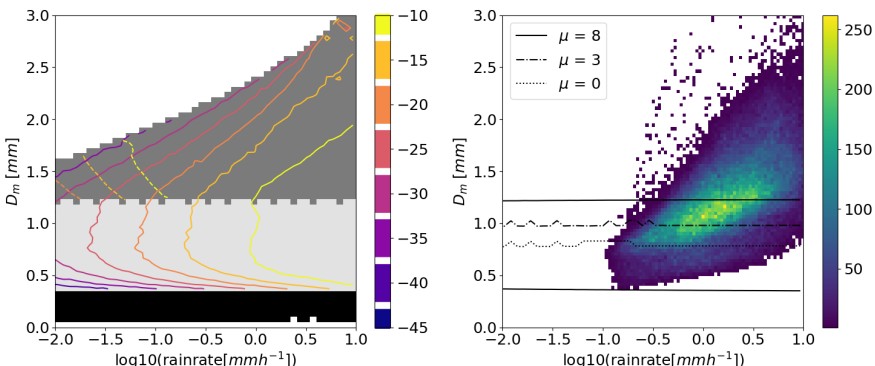

**Figure 3.** The left panel shows the sensitivity required to observe a Mie notch for W-band and G-band radars. The required sensitivity is shown in contours [dBZ], in solid lines for G-band and dashed lines for W-band. The light grey filled region is the region where a G-band Mie notch can be observed but a W-band notch cannot, the dark grey filled region is where both W-band and G-band can observe a Mie notch, the black region is where no Mie notches can be detected. The white filled regions are outside of where PSDs could be constructed with the parameters used. The right hand panel shows the frequency of disdrometer observations within this $D_m$-rainrate space. The black lines represent contours of the region where a G-band Mie notch can be detected but a W-band Mie notch cannot.

Another specific feature of G-band radars over longer wavelengths is the larger amount of attenuation received at G-band at small droplet sizes. This is not desired behaviour, but it is useful for retrieving liquid water content for rain and cloud

profiles where the mean droplet diameter is small and therefore the liquid water content is often also small. Figure 4a shows that while for very small $D_m$s (0.15 mm), the differential extinction coefficient between Ka- and G-bands is already around $10\,\mathrm{dBkm}^{-1}(\mathrm{gm}^{-3})^{-1}$ larger than that of the Ka-W pairing, this gap quickly grows to around $20\,\mathrm{dBkm}^{-1}(\mathrm{gm}^{-3})^{-1}$ for a $D_m$ of 0.5 mm. This increase is similar, though smaller, in the W-G pairing, where the differential extinction coefficient peaks at around $21\,\mathrm{dBkm}^{-1}(\mathrm{gm}^{-3})^{-1}$ at a $D_m$ of 0.4 mm.

This strong increase at such small $D_m$ is due to the fact that attenuation increases strongly in the non-Rayleigh regime to a maximum at $r/\lambda \approx 1.5$ (where $r$ is droplet radius and $\lambda$ is radar wavelength) as suggested in Battaglia et al. (2014). For droplets large enough to scatter geometrically rather than in either the Rayleigh or resonance regimes, the attenuation reduces and there is a very weak wavelength dependence to the attenuation. For instance for the W-G pair, for large $D_m$s the differential attenuation becomes slightly negative (i.e. W-band is attenuated more strongly than G-band).

The differential Doppler Velocity (DDV) can provide an alternative to other methods of retrieving the characteristic diameter due to the simplicity and computational inexpensiveness of the method and the fact that it is not impacted by any reflectivity calibration errors or by the presence of any vertical velocity. This method does, however, require accurate volume matching and excellent vertical pointing calibration so that there is no impact from the horizontal wind on the Doppler velocity. Again, including G-band here improves the estimation of smaller characteristic diameters. Figure 4b shows that the signal for DDV

from Ka- and W-bands is strongest at a characteristic diameter of around 1.8 mm and for diameters less than around 0.4 mm



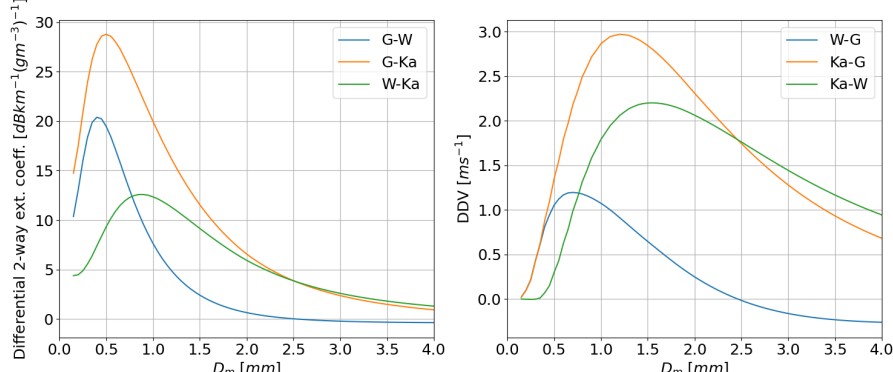

**Figure 4.** Left panel shows differential 2-way extinction coefficient for three wavelength pairing for different sizes of $D_m$. Right panel shows differential Doppler Velocity for three frequency pairings for different sizes of $D_m$. For both panels spectra were generated using an exponential DSD.

the signal is negligible. Compared to this the DDV between both Ka- and G-bands and W- and G-bands give a signal at much lower diameters, reaching down to the lowest $D_m$ considered here. Further the signal from the Ka-G combination is stronger than that of DDV Ka-W, meaning the retrieval will be more reliable for all diameters smaller than around $1.8\,\mathrm{mm}$. There is, of course, a double solution for both W-G and Ka-G combinations, these double solutions can easily be mitigated by using the

DDV in combination with another frequency pairing or with the Doppler velocity itself. In most cases the Doppler velocity should be sufficient to distinguish between the two possible solutions.

## 3   Data and methods

### 3.1   Data

The data used in this study were collected on the 25th May 2021, this is the same as the case study presented in Courtier et al.

(2022). The data were collected during a precipitation event, including periods of both light and moderate precipitations (i.e. rain rates varying between 0.5 and $5\,\mathrm{mm\,h^{-1}}$ according to disdrometer measurements). Doppler spectra and radar moments were collected at Ka-, W-, and G-bands. The radars used here were all located at Chilbolton Atmospheric Observatory and are located within close proximity to one another. The Ka- and W-band radars are in the same location while the G-band is around $20\,\mathrm{m}$ away. The details of the Ka-band, W-band and G-band radars can be found in Courtier et al. (2022).

There was a slight mispointing error found between the G-band radar and the other two radars, this was estimated to be no more than $0.2°$. While there is likely to be some impact from this mispointing, the case studies presented in this study were chosen at times when the standard deviation of the Doppler velocity was low and the horizontal wind was light, in order to attempt to minimize the impacts of both the mispointing and the distance between the radars, though the mispointing did still have an impact on some retrievals.



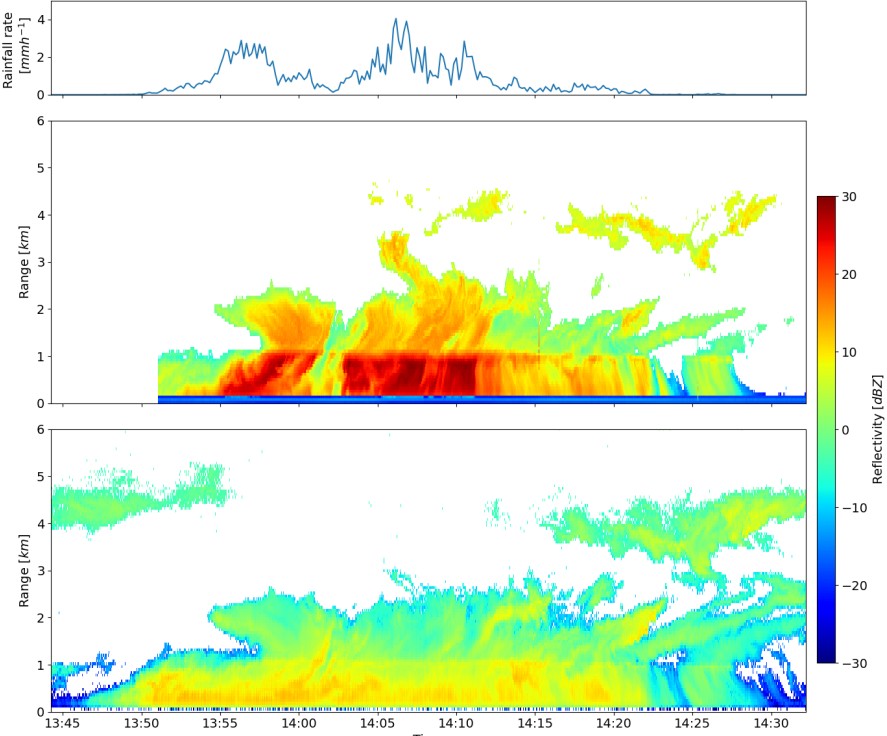

**Figure 5.** Overview of the 2021-05-24 case study. Rainfall rate (a), Ka-band radar reflectivity (b) and G-band radar reflectivity (c).

Figure 5 shows the reflectivity for Ka-band and G-band radars with rainfall rate for the case study. It can be seen that the precipitation rate is generally light, regularly below $4\,\mathrm{mm\,h^{-1}}$. These lighter precipitation rates are more favorable for scanning with the G-band radar as this minimizes the problem of attenuation. Unfortunately, the G-band is attenuated strongly by water vapour. With a surface temperature of $11\,^{\circ}\mathrm{C}$ and high relative humidity, in the lowest kilometer the total 2-way attenuation from all atmospheric gases was $4.9\,\mathrm{dB}$.

## 3.2 Retrievals

### 3.2.1 Vertical wind retrieval

The vertical wind retrieval uses the Mie notch method; this method finds the Mie minima in the measured spectra and then compares their locations with the theoretical predictions. The difference between the observations and the theoretical values provides the vertical wind speed. The application of this method requires some extra modifications to account for the noisiness of the spectra. As such a series of thresholds are used to remove bad data. The first is a prominence (i.e. the difference between the value of the minima and the nearest local maxima) of the minima of $3\,\mathrm{dB}$ or greater; this removes many of the local minima generated by noise. A maximum vertical wind speed of $1.5\,\mathrm{m\,s^{-1}}$ is enforced; in the frontally driven cloud and precipitation





being studied here, this is a reasonable assumption. These thresholds were sufficient to remove much of the noise from the vertical wind retrievals. However, given there are two Mie notches for the G-band an extra constraint can be used in cases where the two Mie notches are visible. That is that the vertical wind speed retrieved from the position of the two individual Mie notches must not differ by more than $0.3\,\mathrm{m\,s^{-1}}$.

### 3.2.2 OE retrieval

The path integrated attenuation (PIA) can be retrieved by using the method of disentangling non-Rayleigh effects and attenuation implemented by Tridon et al. (2013). This method aligns the Rayleigh regions of each frequency spectra and records the vertical shift in dB that the spectra must be shifted to achieve a perfect alignment. This dB value corresponds to the differential attenuation between the two frequencies. However, there were several reasons as to why this could not be implemented here. The assumption that the radars are observing the same targets, used in this method, presents a certain challenge in this case due to some small mispointing errors and the fact that the G-band radar was situated around $30\,\mathrm{m}$ away from the other two radars. Another reason the Tridon et al. (2013) method could not be used was because the sensitivity of the GRaCE radar is somewhat poor. This meant that the Rayleigh region (which only extends to $1.49\,\mathrm{m\,s^{-1}}$ for $200\,\mathrm{GHz}$) was often obscured, either partially or fully, by noise. This meant that matching the Rayleigh regions of the G-band with any other frequency was difficult.

Instead an Optimal Estimation (OE) method was used to retrieve the DSD and therefore the PIA. The methods used for optimal estimation and microphysical retrievals are well established. Firda et al. (1999) first used optimal estimation to retrieve binned DSD using multi-frequency radar observations. Since then, optimal estimation techniques have been widely used in microphysical retrievals (e.g. Hogan, 2007; Tridon and Battaglia, 2015; Mason et al., 2017). The OE method used to retrieve the DSD within this study is implemented as a single level retrieval which retrieves the DSD at a certain level in isolation, and a full column retrieval which uses the DSD retrieved at each level to produce a consistent profile of attenuation. In the full column version the shape of the DSD is retrieved separately at each level but the intensity of the DSDs are adjusted together across the entire column to produce a consistent profile of attenuation.

The OE method is based on that of Mróz et al. (2021) and Tridon and Battaglia (2015). In order to study the effect of G-band on DSD retrievals an OE technique using a combination of Ka-, W- and G-bands is used to compare against a control technique using just Ka-, and W-bands.

Other than the inclusion of the extra frequency (and the benefits derived from that e.g. more frequent measurements of vertical wind) the optimal estimation techniques are equivalent. In each the measurement vector consists of the triple frequency spectral data and the state vector consists of the discretised DSD, together with vertical wind speed and turbulence (both parameters which modify the forward modelled spectra). The a priori of the DSD is calculated from each frequency spectra using a no turbulence assumption, the a priori of the vertical wind speed is calculated using the vertical wind speed retrieval described above, and the a priori of the turbulence parameter is assumed to be $0\,\mathrm{m\,s^{-1}}$. Because there were some small issues with mispointing, as mentioned previously, some corrections for differences in vertical wind speed between the radars were conducted before the optimal estimation. This effect was worsened within the boundary layer as the turbulent nature of the boundary layer makes any influence of the horizontal wind on the retrieval of the vertical wind inconsistent between radars.





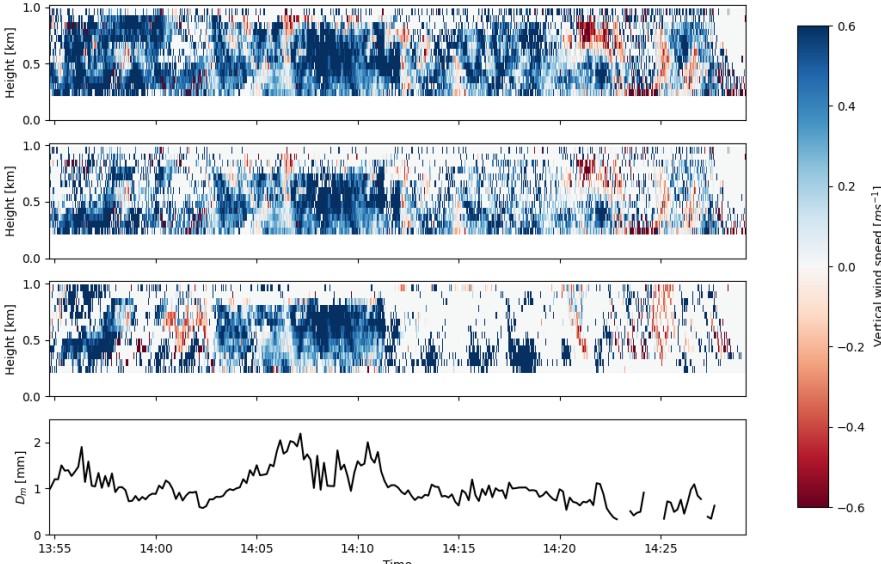

**Figure 6.** Colors show the retrieved wind speed using the Mie notch technique with (a) G-band radar, (b) G-band radar, requiring the first two Mie minima, and (c) W-band radar. (d) shows the $D_m$ time series retrieved from disdrometer observations.

For the full column retrieval, the measurement vector consists of the triple frequency spectral data at each level (in this study, this is nine levels between $450\,\mathrm{m}$ and $950\,\mathrm{m}$ which is the portion of the liquid cloud which has reliable signal from all three frequencies). The state vector consists of the discretised DSD at each level and vertical wind speed and turbulent broadening parameters at each level. The vertical wind speed and turbulent broadening were included at each level (as opposed to a single value for each) due to the highly variable nature of turbulence in the boundary layer.

## 4 Results and discussion

### 4.1 Vertical wind retrieval

One of the vital improvements seen with G-band radar in Courtier et al. (2022) is in the retrieval of vertical wind speed. Because there are no in-situ observations of vertical wind taken simultaneously with the radar observations there is no "truth" to compare against. However, there is a clear improvement shown in Figure 6 in the coverage of wind retrieval. The number of range gates that successfully retrieve the vertical wind at W-band is 41.7% of gates below the melting layer as shown in 6c, while at G-band it is 76.1%. The number of points where it is possible to retrieve vertical wind at W-band is highest near to the ground and in the areas of greatest reflectivity (i.e. mainly greater than $20\,\mathrm{dBZ}$). The coverage from the G-band is much more even and can sometimes extend into the regions of reflectivity smaller than $10\,\mathrm{dBZ}$. At W-band, the number of vertical wind retrievals reduces rapidly after 14:12 when the $D_m$ suddenly drops from around $2\,\mathrm{mm}$ to around $1\,\mathrm{mm}$. Similarly the drop in





$D_m$ from around 13:58 to 14:00 is matched by a reduction in number of vertical wind retrieval points at W-band. This follows what was predicted by theory in Figure 3 where the vertical wind retrieval at W-band is unlikely to succeed at a $D_m$ lower than $1.2\,\mathrm{mm}$, though this has some dependency on the noisiness of the data and the shape of the drop size distribution. As discussed in Section 2, this increased coverage (and therefore reliability) in the vertical wind retrieval will help improve the retrieval of other, more complex, microphysical parameters such as the drop size distribution.

It can be seen in Figure 6b that the number of points using two G-band Mie notches is (as expected) smaller than the overall number of retrieved points. The difference in the number of retrieved points is largely toward the top of this liquid portion of the cloud. This is due to the large amount of attenuation suffered by the G-band, which quickly prevents the second G-band Mie notch from being well resolved compared to the noise. This is also why the two Mie notch retrieval misses some of the points that the W-band does manage to retrieve towards the top of this layer. While it cannot be directly demonstrated here, it is likely that those points at which the spectra has two Mie notches in the G-band are more accurate than the retrievals which only use a single Mie notch, the uncertainty will be reduced by a factor of $\sqrt{2}$. The presence of noise in the Mie notches makes an accurate retrieval of the exact minima difficult and having two minima helps to reduce the errors associated with this noise. Further the two Mie notches improve the reliability of the retrieval technique; as the spectra are noisy it is possible, no matter how rigorous the procedures in place, to mistake a spurious, noisy minimum for a genuine Mie minimum. Where both the first two Mie notches are included within the retrieval the fact that the distance between them is known greatly improves the ability to filter out bad data.

### 4.2 DSD retrieval

Figure 7 shows the Optimal Estimation of the DSD using triple-frequency radar observations in (a) and dual-frequency (no G-band included) radar observations in (b). A major impact from not including the G-band observations in this case is the lack of ability to retrieve the vertical wind speed, this reduces how constrained that parameter is, thereby increasing the possibility of the optimal estimation converging to an incorrect solution. For this specific example there is a $0.82\,\mathrm{m\,s^{-1}}$ difference in the retrieved wind speed between the two methods, this is enough to result in substantial differences in the retrieved rainfall rate and $D_m$.

It can be seen in Figure 7 that there are two areas of large differences in the retrieved DSD, the first is in the smallest particles, where the retrieval without the G-band has a sudden drop in the retrieved number concentration of droplets less than around $0.3\,\mathrm{mm}$. This is a direct consequence of the vertical wind not being retrieved in the no G-band method moving the spectra to the right on the panel 7c, thereby reducing the spectral reflectivity at a smaller Doppler velocities. Further, the non-Rayleigh scattering shown in the simulated spectra in Figure 1 adds extra information about the size of the droplets that allows the OE retrieval of the DSD to better characterise the smallest particles. This non-Rayleigh scattering (a departure from the Rayleigh counterpart of more than $3\,\mathrm{dB}$ in backscatter cross section) begins for droplets with a diameter of $0.37\,\mathrm{mm}$ and brings a strong constraint for the retrieval of the PSD at those smaller sizes. This can be seen in Figure 7 where the OE-retrieved PSD for small particles jumps up and down based on spectral noise.





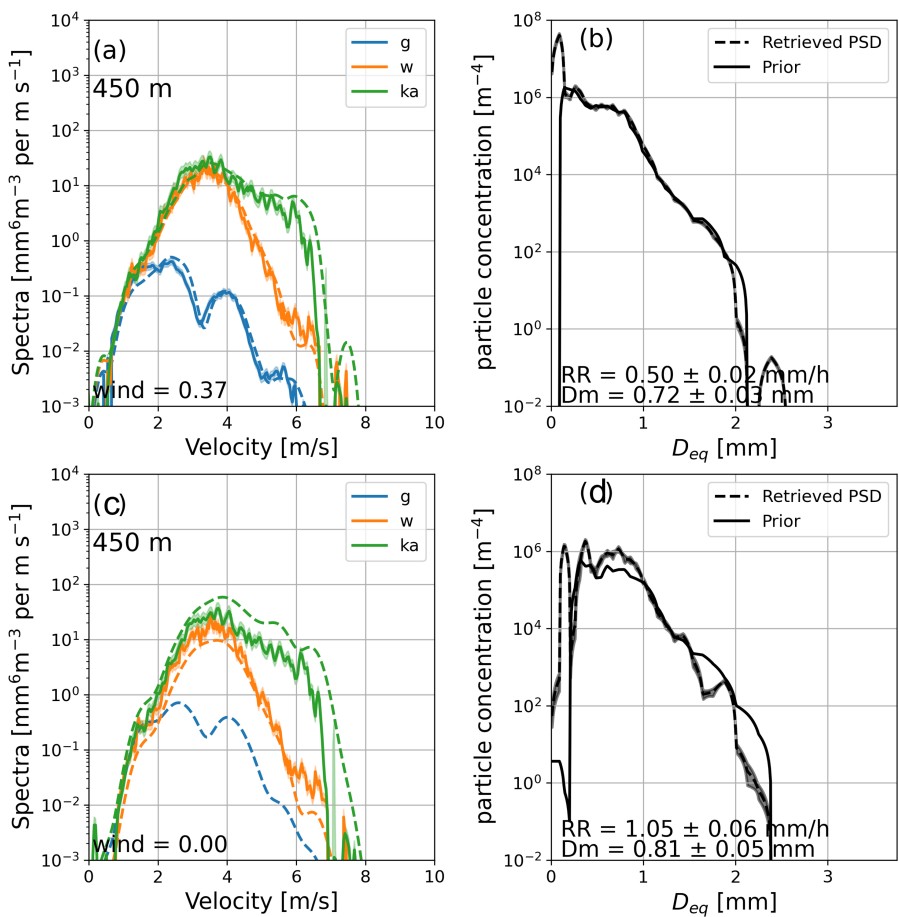

**Figure 7.** Optimal estimation of DSD using Ka-, W- and G-band radars (top row) and Ka- and W-band radars only (bottom row). Left hand column shows observed (solid lines) and forward modelled (dashed lines) radar Doppler spectra at a height of 450m, right hand column shows retrieved DSD.

The second region at which the methods differ is at drop diameters greater than $2\,\text{mm}$. The increase in these larger drops in the no G-band method is again due mainly to the difference in the vertical wind retrieval. There is an increase in the spectral power at velocities greater than $6\,\text{m s}^{-1}$ for the no G-band retrieval as compared to the G-band retrieval. Further, in Figure 7c the small, spurious, increase in the Ka-band spectral power at velocities between $5\,\text{m s}^{-1}$ and $6\,\text{m s}^{-1}$ is caused by the optimal estimation attempting to fill in the Mie notch that would be occurring in the W-band in this region. This creates an overestimation of the drops around $2\,\text{mm}$ in diameter. This in turn helps to increase the value of the $D_m$ beyond just the effects of the vertical wind retrieval alone.



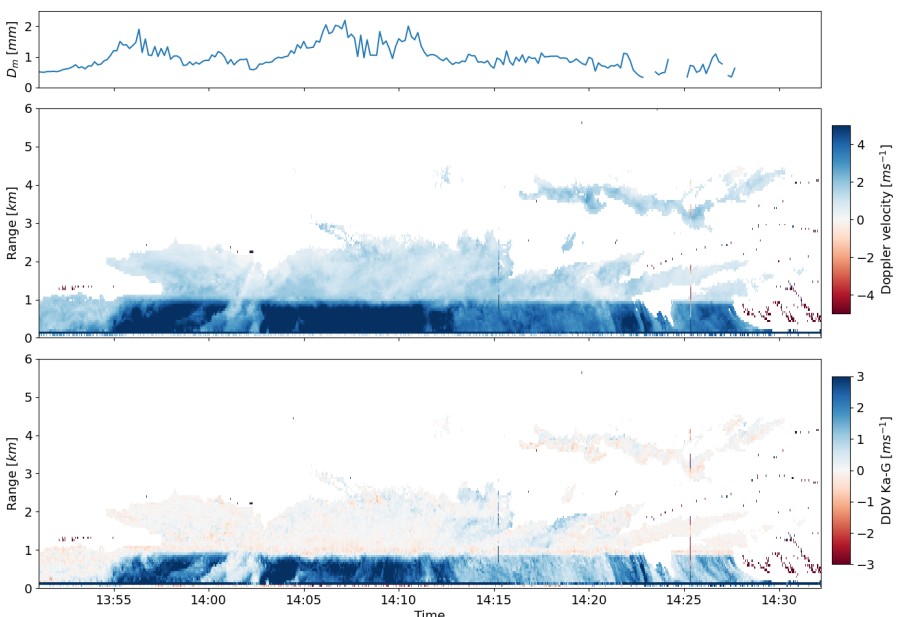

**Figure 8.** Observations of (a) $D_m$, (b) Ka-band Doppler velocity and (c) Ka-G DDV

### 4.3 DDV based retrieval

The improvements that G-band can make to the differential Doppler Velocity are clearly shown in Figure 8. The Ka-band Doppler velocity is shown in the top panel and the DDV between Ka- and G-bands is shown in the bottom panel. The DDV here is very large and the maximum values reaching almost $4\,\mathrm{m\,s^{-1}}$ are equivalent to the peak for a gamma distribution with $\mu = 0$ where the maximum occurs at a $D_m$ of slightly over $1\,\mathrm{mm}$. The DDV in the ice cloud above the melting layer is consistently close to $0\,\mathrm{m\,s^{-1}}$ as is expected in ice and snow where both the particle fall speeds are smaller and the G-band is impacted less by non-Rayleigh effects.

The DDV values taken close to the ground (in order to best match the disdrometer observations) are compared to the observations of $D_m$ in Figure 9a. They are plotted against the theoretical curves of $D_m$ versus DDV assuming gamma distributions with a $\mu$ of 0, 2 and 6. It can be seen that the observations are scattered through the region predicted by theory at large $D_m$ and DDV, but for $D_m$ between about 0.5 and $1\,\mathrm{mm}$ the observations generally underestimate the DDV expected from theory. This underestimation could be due to the inability of the disdrometer to measure droplets smaller than $0.3\,\mathrm{mm}$ which will skew the mass mean diameter to larger values than should be measured.

To investigate this a simple $D_m$ retrieval was used based on using lookup tables generated from the NASA GPM disdrometer observation network. The lookup tables related the $D_m$ measured by the disdrometers with the DDV calculated from forward modelled Doppler spectra based on the DSDs observed by the disdrometers.





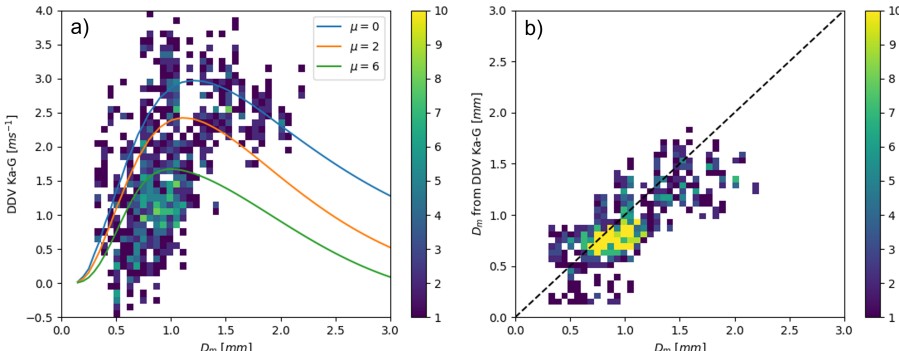

**Figure 9.** a) shows a density plot of observed DDV between Ka- and G-bands and $D_m$ as retrieved from the Jos-Waldvogel impact disdrometer, overplotted are theoretical curves for the relationship between DDV and $D_m$ using a gamma distribution and the values of $\mu$ shown. DDV taken from the lowest usable bin from the radar observations. b) shows a density plot of the retrieved $D_m$ from the DDV observations and the observed $D_m$.

When the $D_m$ is retrieved from the Ka-G DDV observations using the lookup tables, there is a much better fit between the radar observations and the disdrometer observations (see Fig. 9b). This is likely because the retrieval is based on disdrometer
observations (though these are largely 2-DVD disdrometers, rather than impact disdrometers). Williams et al. (2000) show that there is good agreement between measurements of 2-DVD disdrometers and the Joss-Waldvogel impact disdrometer for low DSDs with a low $D_m$, while for larger $D_m$'s this relation breaks down and the impact disdrometer measures the $D_m$ to be larger than the 2-DVD disdrometer. This relationship is also shown in Figure 9b where the radar-based retrieved observations of $D_m$ are smaller than the JWD-based observations. It is, therefore, considered likely that the overestimation of $D_m$ by the
disdrometer is the cause for the theoretical curves in Figure 9a not fitting well for small values of $D_m$. This also raises a potential issue with the use of disdrometers or disdrometer-based retrieval methods for DSDs with a small $D_m$. Because the theoretical curves converge for small $D_m$, it is likely that, below a $D_m$ of around $0.7\,\mathrm{mm}$ a retrieval based on theoretical gamma distributions (with any reasonable value of $\mu$) will be an improvement on disdrometer-based retrievals.

### 4.4 Retrieval of extensive DSD properties

After retrieving intensive quantities (i.e. factors affecting the shape of the DSD such as $D_m$ or $\mu$), extensive quantities (i.e. factors dependent on the intensity of the DSD) such as rainfall rate or LWC can be retrieved. One method of retrieving the LWC involves using the differential attenuation between two frequencies (Hogan et al., 2005).

Here the retrieval of path integrated attenuation and then liquid water path is shown with the inclusion of G-band radar. Because the G-band radar is strongly attenuated it often does not see through to the top of ice clouds and so the common
methods of comparing the reflectivity of small ice crystals between frequencies or matching the Rayleigh regions of spectra (Tridon et al., 2013) cannot be used to retrieve differential attenuation. Instead the differential attenuation induced by the rain between $450\,\mathrm{m}$ and $950\,\mathrm{m}$ (Fig. 10) is computed using the method described in Section 3. At each range gate the attenuation is





calculated based on the simulated PSD. The two case studies show the added value of the inclusion of G-band for PSDs with low $D_m$. In the first case (top row) the $D_m$ is below $1\,\mathrm{mm}$ through most of the considered column (just surpassing $1\,\mathrm{mm}$ in

the last range gate). With these sizes the G-band attenuation is considerably greater than the attenuation at W-band, at times, even more than twice the attenuation at W-band. The second case (bottom row) shows an example with larger characteristic diameters; here the added value from the G-band is smaller, the attenuation estimated at G-band and W-band is very similar, though the attenuation at G-band is still consistently greater than that at W-band.

In the first case (Fig. 10a), the stronger differential attenuation at Ka-G across this small, $500\,\mathrm{m}$, layer enables the retrieval

of LWC with greater accuracy as compared to the Ka-W combination. With an average $D_m$ of $0.86\,\mathrm{mm}$ and an average LWC of $0.23\,\mathrm{g\,m^{-3}}$ across the layer the two-way differential attenuation across the $500\,\mathrm{m}$ layer is $5.2\,\mathrm{dB/km}$ for the Ka-G combination whereas it is $3.1\,\mathrm{dB/km}$ for the Ka-W combination, this is in line with the expectations for each frequency combination shown in Figure 4a, where the Ka-G combination should have a differential attenuation just less than twice that of the Ka-W combination. This means that for DSDs such as this, consisting of small droplets, the Ka-G combination results

in an accuracy almost 2 times more accurate than the Ka-W combination.

In the second case (Fig. 10b), the average differential attenuation for both the Ka-G and Ka-W combinations is larger across this very shallow layer. The average $D_m$ is $1.70\,\mathrm{mm}$ and the average LWC is $0.54\,\mathrm{g\,m^{-3}}$ for this case. This equates to a two-way differential attenuation of $5.7\,\mathrm{dB/km}$ for Ka-G and and a two-way differential attenuation of $3.8\,\mathrm{dB/km}$ for Ka-W. While the Ka-G and Ka-W measurements are more similar than in the small $D_m$ case there is still an increase in the differential

attenuation while using the G-band (as can also be seen in Fig. 4a), again giving the Ka-G combination the ability to be more accurate for profiling LWC in the atmosphere.

In both examples, the differential attenuation matches the expectations from the theory presented in Section 2. For the first example the differential attenuation between G-band and W-band is large, while the LWC for each range gate is small. This leads to small values of total attenuation even in the G-band. For the second example the observations again match theory

closely in that the two-way differential attenuation between W- and G-bands is significantly reduced. This should be the case as the G-W differential attenuation is significantly reduced by a $D_m$ of $1\,\mathrm{mm}$ and becomes negative by $1.5\,\mathrm{mm}$. However, because of the greater amount of liquid water in the column the overall value of the average differential attenuation in the layer is still larger than that of the small $D_m$ case.

## 5 Conclusions

G-band radars can provide substantial extra information in the rain microphysical characterization both through the non-Rayleigh scattering from small droplets and the associated Mie notches, and the increased attenuation experienced at this frequency. The added value of using the G-band in combination with other frequencies is demonstrated through a number of retrieval methods. The improvement in the vertical wind retrieval has a solid foundation: the two Mie notches occur in the $200\,\mathrm{GHz}$ G-band spectra in correspondence to raindrops with sizes smaller than for the Mie notch in W-band spectra.

This means that the vertical wind can be retrieved from G-band spectra at much lower rainfall rates and droplet diameters



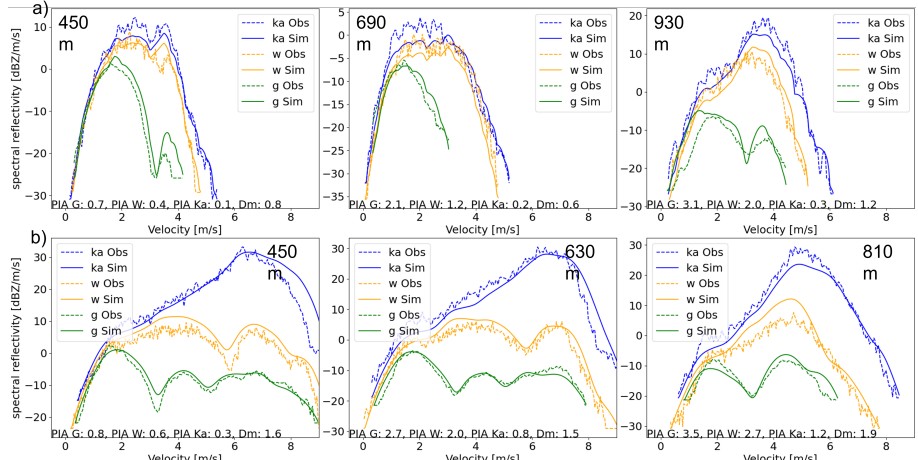

**Figure 10.** Observed (dashed lines) and simulated (solid lines) spectra at three levels within the liquid portion of the cloud. Attenuation, $D_m$, and LWC calculated from the simulated PSD are annotated for each range gate shown. Top row shows a case at 14:01:00 with a small $D_m$, bottom row shows a case at 14:07:00 with a large $D_m$.

than by using W-band, thus enabling retrievals of the vertical wind to be extended to regimes of smaller raindrop size and typically smaller rain rates. Moreover, the presence in some spectra of two Mie notches mean that there is increased precision and certainty in the value of the vertical wind that has been retrieved. This is especially important for turbulence broadened spectra or for noisy spectra where the exact location of the Mie minima may be more uncertain. The improvement in the DSD

retrieval is more pronounced for the smallest droplets (i.e. for droplets with diameters less than $0.5\,\mathrm{mm}$). This has an impact on the values of the rainfall rate and the mass-mean droplet diameter calculated from the DSD. For the retrieval of mass-mean diameter using the DDV method, the G-band adds considerable value for the smallest values of $D_m$; this is especially important as these are the cases where disdrometers have troubles at retrieving an accurate value for the mass-mean diameter. The increased dynamic range for the Ka-G DDV pairing reduces the uncertainty in the retrieval of the $D_m$, for a given error in

the Doppler measurements.

There is a similar enhancement in the dynamic range of differential attenuation. Compared to W band, the increase in the value of differential attenuation for large diameters is relatively small. However, for small rainfall rates and, in particular, low values of LWC, the improvement in the differential attenuation between Ka- and G- bands as compared to Ka- and W-bands is very impactful and allows for the reliable retrieval of LWC to much lower values of LWC. Overall, the use of G-band radar

in appropriate environments (e.g. cold and dry air) has the potential to enable more accurate retrievals of LWC, DSDs and $D_m$ and to extend such retrievals to regime of drizzle and low rain rates.





*Author contributions.* BC wrote the paper and conducted the analysis. AB supervised the data analysis. AB and KM contributed to the retrieval methods and analysis. AB and KM contributed to the paper.

*Competing interests.* The authors declare that they have no conflict of interest.

*Acknowledgements.* The work done by B.M. Courtier and A. Battaglia was funded by the UK NERC project GRACES (G-band RAdar for Cloud and prEcipitation Studies, grant number RP16G1219). The work by K. Mroz was performed at the University of Leicester under grant
no. RP1890005 with the National Centre for Earth Observation.



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
