# Peer review of "Advantages of G-band radar in multi-frequency, liquid phase microphysical retrievals"

_EGUsphere, 2024_

## Author Comment (AC1)

We thank the reviewer for their attention to the manuscript and the comments they have made. We reply to these comments below (in blue).

General comments:

My main concerns arise from your frequent statements that the G-Band "improves clearly" the retrieval of various microphysical properties, while you are most of the time not showing any comparisons with in-situ observations or other independent measurements or you are not showing the retrieved quantity from both the Ka-G and Ka-W combination.

While I do agree that the G-Band likely improves the accuracy of the retrieval of the drop size distribution (DSD), the example you provide in the paper in my opinion does not show that the inclusion of the G-Band actually improves the accuracy of the retrieved DSD. Yes, the forward simulated spectra look more similar when you include the G-Band, however, that does not mean that the retrieved DSD is actually "more correct". You have DSD measurements from the disdrometer, could you compare the DSD measurements with the retrievals? Or perhaps you could compare the forward simulated Doppler spectrum to the Doppler spectrum of a fourth, independent radar at a different wavelength which is not used in the retrieval if you have access to another radar.

The disdrometer observations are difficult to use as a ground truth in this case. There are several issues with the disdrometer: it has a poor sampling resolution compared to the radar, in order to get a reasonable count it must collect droplets over a length of time typically at least an order of magnitude longer than the averaging time of the radar spectra and most importantly the disdrometer is measuring the DSD at the surface while the radar observations are taken at 450m, this means that even is some adjustment is made based on average drop velocity the disdrometer and the radar are not observing the same thing. There are also unfortunately no other radars to compare against in this situation.

We have tried to address the issue of comparison, (in line with Reviewer 2's comments) we have made a more robust inclusion of OE error in the manuscript, also showing the benefit of including the G-band in reducing the error.

Also, in your example using DDV to retrieve Dm, you are showing a comparison of Dm retrieved and Dm measured by the disdrometer, however, here I am missing a comparison to the retrieved Dm from the Ka-W DDV to actually show any improvements in comparison with the lower frequency pair. So my main point is: I do not think that you have shown clearly that the addition of the G-Band radar

improves the retrieval of the DSD (or Dm), you have just shown that it can retrieve a DSD (which does not need to be accurate) and that the Dm you retrieve with the G-Band is rather accurate, however, you don't show that the one retrieved from Ka-W is less accurate. I think the paper would benefit greatly if you include a more detailed comparison of the DSD with in-situ (or other) observations and if you compare the retrieved Dm not only with in-situ observations but also with the one retrieved from Ka-W.

 We have now included the retrieval of Dm with the Ka-W band DDV combination in Figure 9 and some discussion comparing the Ka-W with the Ka-G

Specific comments:

Line 32: what is the smallest D0 you can retrieve with W-Band? Please specify in the text. This is about 0.7 mm, this has been added to the text

Line 47: please specify what you mean with DDV (probably dual-doppler velocity) Clarified in text

Line 52: perhaps you could elaborate on why there may be double solutions for the DDV retrieval using two radar wavelengths? An explanation has been added in the revised manuscript

Line 65: Do you mean above the freezing level? Below the freezing level ice can cause significant attenuation. Or do you mean below the freezing level in respect to the height? Then you should clarify that. Clarified in text that it is below the freezing level with respect to height

Line 80: remove "to this radiation" Done as suggested

Line 112: remove the second "the sensitivity" Done as suggested

Line 114-115: could you elaborate more on why there are small variations in the success of the G-Band detecting a Mie-notch? Done as suggested

Line 175: possibly "bad" data should be replaced with "unsuited" data Done

Line 177: why do you restrict the vertical wind speed to a maximum of 1.5m/s? In this situation the vertical wind speed should not be greater than that and introducing a reasonable limit to the wind speed decreases the possibility of spurious minima being treated as a Mie notch

Line 182: Perhaps it would be better to call this section "Optimal estimation retrieval", since OE has not been introduced yet Changed as suggested

Line 188: is the G-Band 20 or 30 m away from the others? In the Methods section you said 20 m, here 30 it is about 30m, the method section has been changed

Line 243: replace bad with another adjective Done as suggested

Figure 7c: why is the forward simulated W-band spectrum fitting so badly to the observed spectrum? Is the OE not working properly? We have changed the OE slightly for reviewer 2's comments, the new OE fits better, but part of the OE is fitting as best as it can to all three spectra, this means that it will miss some parts on each

Line 268: I do not see clearly how the G-Band improves the differential Doppler velocities. Perhaps it would be helpful to show Ka-W in addition to Ka-G? Done as suggested

Line 273: why is the G-Band affected less by non-Rayleigh scattering in the ice phase? Ice particles do grow rather large and cause significant non-Rayleigh scattering already in W-Band right? Clarified that this is due to the smaller absolute difference between the Doppler velocity in a non-Rayleigh scattering regime and Rayleigh scattering regime for ice.

Figure 9: why do you not show the Dm retrieved from Ka-W? I thought that was the whole point of the paper, to show the benefits the addition of the G-Band could have compared to just having lower frequency radars Done

Line 284: Much better fit compared to what? Compared to the theoretical curves and the scatter plots of DDV obs vs disdrometer obs. This has been added to the text

Line 303 and following: is there a way you can actually show that the LWC retrieved from the G-Band is more accurate than from the W-Band? I agree that it has a larger potential because the absolute values of PIA are larger, but does that actually make such a big difference? Added sentence in intro for reduced error with Ka-G combo can't actually verify LWC for real world example

Figure 10: could you please add units to the PIA and Dm in the plots? Done as suggested

All your Figures: it is probably a matter of taste, but I would rather have a label on the colorbar than having to search the figure caption for the description of what is plotted here. So I would suggest that you include colorbar labels Done as suggested

---

## Author Comment (AC2)

The manuscript illustrates the advantage of including G-band radar measurements at 200GHz to retrievals of vertical wind speed and droplet size distribution (DSD) parameters. The authors describe the theoretically expected potential of including G-band measurements; introduce three different retrieval approaches (vertical wind; optimal estimation technique for DSD; Dm through differential Doppler velocity DDV), and evaluate the advantages based on one test case previously described in Courtier et al, 2022. While the presented case study of G-band measurements offers a lot of exciting material for the different retrievals presented, the main messages of the paper need to be strengthened to underline the advantages of the G-band.

**General comments:**

-GC1: The authors mention often throughout the text that including the G-band to their retrievals improves the retrievals compared to the KaW combination. Yet, I do not think that this message is underlined enough by the presented analysis and choice of figures. I would suggest two things: i) retrieved results including G-band should be compared in more depth to retrieved results using only Ka or W-band; ii) independent measurements should be taken into account to serve as "truth". If independent measurements are not directly available, retrieval results using the different radars could also be compared to each other in forward simulated radar space.

The disdrometer observations are difficult to use as a ground truth in this case. There are several issues with the disdrometer: it has a poor sampling resolution compared to the radar, in order to get a reasonable count it must collect droplets over a length of time typically at least an order of magnitude longer than the averaging time of the radar spectra and most importantly the disdrometer is measuring the DSD at the surface while the radar observations are taken at 450m, this means that even is some adjustment is made based on average drop velocity the disrometer and the radar are not observing the same thing. There are also unfortunately no other radars to compare against in this situation.

We have tried to address the issue of comparison through a more robust inclusion of OE error in the manuscript (in line with GC2), showing the benefit of including the G-band in reducing the error. Together with this we have shown a second example of the DSD retrieval where there is less added benefit from the G-band highlighting the value of it where it is most applicable.

-GC2: The authors base some of their results on the very powerful optimal estimation retrieval tool. Yet, not the full potential is exploited in the current analysis. I would suggest to analyse the advantages of including G-band by making use of eg the aposteriori errors and information content, and to compare these to the setup using conventional Ka-/W-band retrieval. These results should be illustrated in additional figures (also see specific comment on Fig 7 below).

We have changed the a priori meaning that the aposteriori errors are more directly comparable. The aposteriori errors have now been included in the manuscript to illustrate the reduction in error that the G-band retrieval has as compared to the W-Ka band retrieval.

-GC3: All analysis is based on a case study with light rain of 45 minutes in total. The different retrievals are applied to different times within the covered measurement phase. In my opinion, the advantage of including the G-band could be highlighted more by using all three retrievals for the same selected time stamp. This 'golden case' could be used as a synthesis bringing the different retrievals and advantage of the G-band together. It would also be interesting to include two different time stamps with different rainfall intensities to illustrate when the retrieval techniques (and advantages) are most or least beneficial.

We have included a large rain rate case, in the DSD retrieval (only the attenuation and DSD retrievals are done at specific times) to show that if the W-band has data on the vertical wind speed then the added value of the G-band is less. We have not matched the two retrievals time stamps as they both display different characteristics better than the other

The vertical wind speed and Dm (from DDV) retrievals are both done across the whole time series

**Specific comments:**

- This might be a matter of taste, but I would encourage the authors to embed subsections 1.1-1.3 in the overall introduction text without subsections. Strengths and drawbacks of the different retrievals introduced here should be sharpened to clarify the motivation for the study. The state-of-the-art for optimal estimation applications to retrieve DSDs needs to be added to the introduction. The subsections have been embedded in the overall introduction and the introduction has been updated

- Sec 3.2: The presentation of the numerous different retrievals with each different inputs would benefit from an overview table summarizing each retrieval's method, input measurement, output retrieved variable, reference to each method. This has been added

- Section titles should be chosen more consistently throughout the manuscript to facilitate the readers' orientation. I would suggest to maintain naming the retrieval sections according to what variable is retrieved by what technique, and to keep the titles consistent between Sec. 3.2 and 4. For example, Sec. 3.2.2 could be renamed to 'DSD retrieval using optimal estimation'. A description of the DDV retrieval method should be added to Sec. 3.2. This has been changed and the DDV section added

- L 162: what observations were used to monitor horizontal winds? At what height levels? The horizontal winds were taken from ECMWF model data and were checked at heights relevant to the observations, i.e. <2km

- Fig 5: a panel showing a flag when case studies were suitable to apply retrievals should be added (L162); and if available, a time line of IWV and maybe LWP, or at least state the IWV in the text (L168) to provide a framework of the stated attenuation. In order to compare Ka, W, and G, it would be nice to add a panel illustrating the W-band measurements for this case, and to add a sub-title to each panel clarifying which frequency is shown. Vertical lines or markers should be added at the DSD case study times chosen for Figs 7 and 10 (or the same time stamp could be picked, see GC3). Why are ice cloud features at 4km height more pronounced in G-band at 14:15 – 14:30, when attenuation is stronger in G-band? The W-band was initially not added as it is similar to the Ka-band and we thought it was better to keep the figure concise, it has now been added. The vertical lines when the retrievals were taken have been added to the Figure. The ice cloud features in the Ka-band subplot were removed accidentally in data quality control, this has been rectified.

- L 205: the text should include information on how the covariances were defined in the optimal estimation retrieval. This has been added

- L232: The authors should clarify in the text where they are pointing at in Fig 6b. This has been added

- L236 ff: This assumption should be underlined with an analysis of the existing data. The authors could show an example by eg zooming in on one time stamp (also see GC 3) to illustrate their hypothesis. We could not do a robust comparison of the accuracy of the vertical wind retrieval as we do not have and observational data to verify against. The statement that the retrieval using two Mie notches will be more accurate refers to the fact that the uncertainty is reduced due to multiple measurements. This has been clarified in text.

- Fig 7 b), d): It is unclear from the figure caption which radar is used for the presented retrieved DSDs. Retrieved DSDs seem to be dominated by the prior assumption, with little information from the observations. As stated in GC2, the potential that OE offers to analyse this case in more depth should be used here to clearly state the information gain by the observations compared to the prior (eg Degrees of Freedom for signal; Averaging Kernel; Jacobian), and benefits on retrieved error thanks to addition of the observations. We have updated the OE so that the prior is the same for both the retrievals with and without the G-band, the prior is now the average DSD of the precipitation event, as retrieved by the disdrometer. The benefits of the G-band to the reduction of error is now discussed in the text.

- Fig 9b): statistical measures of the comparison like RMSE, bias, correlation coefficient would highlight the comparison. The Bias and correlation coefficient have been added as annotations to the Figure and are discussed in text

- Sec 4.4 and Fig 10: I would suggest to replace this figure with a plot showing the retrieved time line of LWC, LWP and rainfall (using Ka/W; and including G); or the retrieved LWC profile for a chosen time stamp (also see GC3) in order to underline the statement given in L298 and the section title. The rainfall retrieval could be evaluated with the independent observations given in Fig 5 top panel and illustrated eg in a scatter plot. We have added the LWC profile to Fig 10 and the rainfall retrieval time series in a separate figure

**Technical details:**

- labels should be added to all colorbars

- keep DSD (instead of PSD) as label for consistency throughout manuscript (eg Fig 7; L258)

- no abbreviations should be used in Section titles (Sec. 3.2.2, 4.2, 4.3)

- readability of the manuscript would benefit from shortening many sentences throughout the manuscript which stretch over multiple lines separated by commas (eg L110, 154, 161, 251).

These details have been corrected as suggested

---

## Referee Report (RR1)

**Review of *Advantages of G-Band radar in multi-frequency, liquid phase microphysical retrievals* Courtie, Battaglia, Mroz**

First I would like to thank the authors for their effort in addressing my comments from the last review. In my opinion, the quality of the paper has improved significantly. I would still like to point out that without independent measurements of the DSD or LWC, some of the claims regarding the increased accuracy of the retrieval can not be made. I would suggest rephrasing those sections (see general and specific comments), or add additional observations/comparisons.

General comments:

1. I understand that the disdrometer you are using has its problems with measuring the DSD accurately, especially at small sizes, however, many radar observationalists (e.g Dias Neto et al. 2019, Myagkov et al. ) are using disdrometers in order to calibrate the radar reflectivity by forward simulating the measured DSD and comparing the simulated reflectivity to the observed reflectivity. Since the radar blind zones are also an issue in this calibration, there of course arise uncertainties with respect to the reflectivity comparison. However, in cases with moderate rainfall in stratiform events, Myagkov et al. have shown that the comparison between disdrometer and radar provide an accurate estimate of radar reflectivity offset of the radar. So, in my opinion, in specific cases the comparison between disdrometer and radar can be possible. Have you tried to compare the retrieved DSD and the observed DSD? Are the measurements of the disdrometer really not usable? Could you at least compare the retrieved rain rates to the measured ones?

2. In the DSD retrieval, you are comparing dual-frequency (Ka-W) to triple-frequency (Ka-W-G) and you say that the inclusion of the G-Band increases the accuracy of the retrieval. If you want to sell the point that the G-Band has an added value, I would suggest to compare dual frequency (Ka-W) to dual-frequency (Ka-G) instead of triple-frequency, because most likely simply the introduction of a third frequency improves the retrieval, regardless of it being a G-Band radar of e.g. a X-Band radar. So I would suggest to include the retrieved DSD using only the Ka-G band as an additional comparison.

3. In the retrieval of the LWC using differential attenuation you claim that the Ka-G has a differential attenuation which is twice that of the Ka-W differential attenuation, therefore the retrieved LWC is twice as accurate. In my opinion you can not make that claim without an independent measurement of the LWC. I understand that the independent measurement is not available for you, but then in my opinion you can not make such a strong claim. I think it is justified to say that the stronger differential attenuation in Ka-G is most likely

increasing the accuracy, but without an independent measurement you can not say that definitively. So, I would suggest if you do not want to or can include other measurements to rephrase the paragraphs and to reduce the claim of two times better accuracy.

Specific comments:
1. Line 38/39: here you already mention differential mean doppler velocity, so you can already introduce DDV here as an acronym
2. Line 47: it should be either Doppler spectra are recorded or Doppler spectrum is recorded
3. Line 52: typing error in Rayleigh
4. Line 68: please introduce PRF
5. Figure 1: what do you mean by arbitrary power? (as the y-axis label)
6. Line 122: missing is: the Mie notch is 0.8mm
7. Line 132: missing a: This is not a desired behaviour
8. Line 161: I would suggest to replace around with approximately
9. Figure 5: what are the dashed lines in the figure? Could you add that to the caption?
10. Line 203: What does GRaCE stand for? you have not used this acronym before
11. Figure 7: Where can I see the reduction in error?m Is that the shaded area around the retrieved DSD? If so could you specify this in the caption? Also, the shaded area is hardly visible, perhaps you could change the line and shading colour of the retrieved DSD. Or is the reduced error the e.g. plus-minus 0.04mm/h you have added to the rain rate and Dm? Because then in my opinion the difference between Ka-W-G and Ka-W is not that large.
12. Line 299-304: Not really clear to me what you are doing/comparing here. Could you write that more clearly? E.g. specify that you are plotting the measured DDV against the from the disdrometer measured Dm in that plot. Also, from which specific height are you taking the DDVs?
13. Line 305: remove using (... retrieval was used based on lookup tables…)
14. Line 316: please introduce the acronym JWD

---

## Author Response (AR2)

**Response to Reviewer**

We would like to thank the reviewer for spending their time reviewing the manuscript again and for their comments. They have been very helpful, in our view, in improving the manuscript and making the study more complete. We include answers to their general comments below. All specific comments were corrected as suggested

1) We agree that it would make the study much more complete to do this comparison, but we do not believe that it is possible given the data quality that we have. Figure S1 attached below show the retrieved DSD for the two case study times in Figure 7 of the manuscript together with the disdrometer observations for the minute surrounding the best estimate of the time of DSD observation (that is the radar time plus 5 minutes). It can be seen that the disdrometer observations do not vary much between the two cases in terms of magnitude of the drop count and the observations do not spread across much of the retrieved DSD. We do not believe that any comparison against the disdrometer observations is a useful comparison for this study. We have moderated some of the language within the text to allow for the lack of independent measurements.

2) We have included the Ka-G retrieval in Figure 7 for the two case study times. Text has been added throughout section 4.2 to discuss the new subfigures.

3) We have rephrased the paragraph as suggested to suggest that there is an improvement but it cannot be verified without independent measurements of LWC.

[Figure]

Figure S1: As Figure 7 within the manuscript but only showing the two triple frequency retrievals. The crosses on the right-hand figures show the disdrometer observations averaged over a 1 minute period centred on the time of the radar observations.